# Identification and evolution of novel *cfr*-carrying plasmids in XDR *Klebsiella pneumoniae* strains from a chicken farm

Yiming Feng,[1] Tiantian Li,[1] Shiyun Zhao,[1] Mengxiang Zheng,[1] Jiaxing Shen,[1] Xiaoying Wu,[1] Xuexue Li,[1] Yajun Zhai,[1,2,3] Li Yuan,[1,2,3] Jianhua Liu,[1] Gongzheng Hu,[1,2,3] Yushan Pan,[1] Dandan He[1,2,3]

**ABSTRACT** The *cfr* gene is typically located on plasmids, playing a crucial role in both intra-species and inter-species transmission among bacteria. This study focused on two *cfr*-positive *Klebsiella pneumoniae* strains, J21CTR26 and J21CTR30, isolated from environmental samples of a chicken farm. Antimicrobial susceptibility testing showed that these strains exhibited an extensively drug-resistant (XDR) phenotype, including resistance to 16 antibiotics across nine classes, notably tigecycline and colistin. Sequence analysis identified both strains as belonging to K10-ST6262, harboring 32 resistance genes and 5 virulence genes. Despite their genetic similarity, hybridization and whole-genome sequencing revealed distinct plasmid profiles. Strain J21CTR26 harbored a 5,404,868 bp chromosome and three plasmids: pCTR26-1 (485,030 bp), pCTR26-2 (141,660 bp) carrying *cfr*, and pCTR26-3 (106,823 bp). In contrast, J21CTR30 possessed a 5,405,166 bp chromosome and three plasmids: pCTR30-1 (394,261 bp) carrying *cfr*, pCTR30-2 (234,624 bp), and pCTR30-3 (106,770 bp). Conjugation experiments demonstrated the transferability of the *cfr*-positive plasmid pCTR26-2 from J21CTR26 to *Escherichia coli* J53, with a conjugation frequency of $4.2 \times 10^{-6}$. Reverse PCR indicated the formation of a circular intermediate, IS*26-cfr*, in both *cfr*-positive strains. Sequence analysis suggested that pCTR30-1 likely originated from a recombination event between pCTR26-1 and pCTR26-2 facilitated by Tn*As1*-mediated homologous recombination. Identification of a fusion plasmid and its daughter plasmids in different XDR *K. pneumoniae* strains underscored the dynamic recombination and spread of resistance plasmids in agricultural environments. In conclusion, this study provided insights into the dissemination and evolution of *cfr*-positive plasmids in husbandry, shedding light on their critical role in antibiotic resistance persistence.

**IMPORTANCE** This study highlights the alarming role of *K. pneumoniae* in the spread of antibiotic resistance within agricultural environments. Two strains isolated from a chicken farm were found to carry the *cfr* gene, conferring resistance to multiple critical antibiotics, including tigecycline and colistin. Despite their genetic similarity, the strains exhibited distinct plasmid structures, emphasizing the complexity of plasmid evolution. The identification of a fusion plasmid and its derived plasmids underscores the dynamic nature of resistance gene transfer facilitated by recombination events. Importantly, the transferability of the *cfr*-positive plasmid to *E. coli* demonstrates the potential for cross-species dissemination. This work sheds light on how agricultural practices contribute to the persistence and evolution of resistance genes, with significant implications for public health and the global fight against antimicrobial resistance.

**KEYWORDS** *Klebsiella pneumoniae*, *cfr*, Fusion plasmid, Tn*As1*

Address correspondence to Yushan Pan, pylearn21@163.com, or Dandan He, hedandan_68@126.com.

Yiming Feng, Tiantian Li, and Shiyun Zhao contributed equally to this article. The author order is determined by descending seniority.

The authors declare no conflict of interest.

The excessive use of antibiotics is considered a major factor in the rapid emergence of extensively drug-resistant (XDR) bacterial strains, prompting bacteria to develop

resistance mechanisms against antibiotics (1). In 2000, Professor Stefan Schwarz of Germany serendipitously isolated a strain of *Staphylococcus aureus* from nasal swabs of calves with respiratory infections. This strain exhibited resistance to tetracycline, erythromycin, kanamycin, chloramphenicol, and florfenicol, and the *cfr* gene was identified within this strain (2). *cfr* modifies the A2503 site on 23S rRNA by the encoded methylase, leading to C-8 methylation of A2503 (m8A2503), which leads to resistance to eight classes of ribosomal-targeted antibiotics (3). In recent years, the *cfr* gene has been detected in isolates from different sources around the world, including gram-positive and gram-negative bacteria, such as *Staphylococcus*, *Bacillus*, *Proteus*, *Enterococcus*, *Escherichia coli*, *Macrococcus*, *Streptococcus*, and *Pasteurella* (4). Since Wang et al. (5) reported *cfr*-positive *Klebsiella pneumoniae* from swine, no other *cfr*-positive *K. pneumoniae* was reported. In this study, for the first time, we found two *cfr*-positive XDR *K. pneumoniae* isolates, J21CTR26 and J21CTR30, from environmental samples at a chicken farm.

Several studies have identified *cfr*-carrying plasmids such as IncFII, IncX4, IncA, IncC, IncN, IncP, and various unidentified types (6–11), while no *cfr*-carrying IncFII(K) plasmids have been reported so far. In this study, we identified two novel *cfr*-positive plasmids characterized by the IncFII(K) plasmid backbone. IncFII(K) is the primary plasmid replicon in *K. pneumoniae* and serves as a major carrier for the dissemination of various resistance genes, including $bla_{NDM-1}$, $bla_{KPC-2}$, $bla_{CTX-M-15/55}$, $bla_{IMP-26}$, *tet*(A) variant, $bla_{TEM-1/141}$, and *fosA3* (12). Plasmids commonly harbor numerous insertion elements, and their variable regions frequently contain antibiotic resistance genes flanked by different insertion sequence (IS) elements. These IS elements play a pivotal role in mediating recombination and transposition events that facilitate the spread of the *cfr* gene, with IS*26* closely associated with the dissemination of *cfr* (9). This study analyzed the evolutionary mechanisms of two novel formed *cfr*-positive plasmid fusions. These findings are crucial for understanding and curbing the spread of drug resistance in agricultural settings.

## MATERIALS AND METHODS

### Sample collection and identification

Environmental samples (chicken house ground and coop) were collected in March 2021 in Jiangxi Province. The samples were incubated in Luria-Bertani broth for 16 to 18 hours at 37°C and then plated on MacConkey agar. Isolates were subsequently purified and identified using the VITEK-2 Compact system (Biomerieux, Marcy l'Etoile, France) and 16S rRNA gene sequencing. A total of 18 *cfr*-positive strains were isolated from 223 strains, and 2 *cfr*-positive *K. pneumoniae* strains were selected for further study.

### Antimicrobial susceptibility testing

The microbroth dilution method was used to detect the minimal inhibitory concentrations of 18 antimicrobial agents for *K. pneumoniae* strains. The 18 tested antimicrobials contained ampicillin, ceftriaxone, cefotaxime, ceftazidime, doxycycline, tigecycline, tetracycline, kanamycin, amikacin, streptomycin, ciprofloxacin, enrofloxacin, linezolid, colistin, meropenem, fosfomycin, lincomycin, and florfenicol. *E. coli* American Type Culture Collection (ATCC) 25922 and *Staphylococcus aureus* ATCC 29213 were used as quality control. The results were interpreted based on the Clinical and Laboratory Standards Institute (13). Breakpoints of tigecycline and florfenicol were interpreted according to the European Committee on Antimicrobial Susceptibility Testing (https://mic.eucast.org/).

## Detection of resistance gene and circular intermediate

The presence of the *cfr* gene and the circular intermediate of IS*26-cfr*-IS*26* was detected using PCR and sequencing as described previously (11).

## S1-PFGE and Southern blotting

The plasmid profiles of J21CTR26 and J21CTR30 were verified by S1-PFGE. The location of the *cfr* gene was identified by Southern hybridization with *cfr* as probe.

## Conjugation assays

The transferability of the *cfr* gene was detected by conjugation test using *K. pneumoniae* J21CTR26 and J21CTR30 as donors and sodium azide-resistant *E. coli* J53 as recipient. Transconjugants were selected on MacConkey agar supplemented with florfenicol (16 mg/L) and sodium azide (150 mg/L). The conjugation frequency was calculated as the number of transconjugants per recipient.

## Whole-genome sequencing and bioinformatics analysis

The genomic DNA of the strain was extracted using the QIAamp DNA Mini Kit (QIA-GEN, Hilden, Germany). Whole-genome sequencing was performed using Illumina Hiseq 2000 and Oxford Nanopore Technologies MinION platforms. Sequencing reads, including short-read and long-read data, were assembled with Unicycler version 0.4.4 with the hybrid assembly strategy. The sequence was initially annotated using the RAST server (http://rast.nmpdr.org) and corrected manually based on the National Center for Biotechnology Information (https://www.ncbi.nlm.nih.gov/). Antimicrobial resistance genes (ARGs) and plasmid replicon types were identified using ResFinder and Plasmid-Finder software on the CGE server (https://cge.cbs.dtu.dk/services/). IS elements were determined using ISfinder (https://isfinder.biotoul.fr/). The serotype of *K. pneumoniae* was determined by wzi allele-based capsule typing (https://bigsdb.pasteur.fr/). The putative coding sequences were identified using the Open Reading Frame (ORF) Finder program (http://www.ncbi.nlm.nih.gov/projects/gorf/). Plasmid profiles were generated using BRIG or Easyfig tool (14, 15).

## RESULTS

### Characteristics of *cfr*-positive *K. pneumoniae* strains

Two *cfr*-positive *K. pneumoniae* strains, J21CTR26 and J21CTR30, were isolated from environmental samples collected from chicken cages and floors within a poultry farm. Antibiotic sensitivity testing showed that both strains exhibited XDR profiles, maintaining susceptibility only to meropenem and streptomycin. They displayed resistance to a broad spectrum of antibiotics, including tigecycline, colistin, and fosfomycin (Table 1). High-level colistin resistance may be caused by mutations in mgrB, pmrB, and crrB. S1-PFGE and Southern blotting results indicated that the *cfr* gene was located on a ~140 kb plasmid in strain J21CTR26 and a 390 kb plasmid in strain J21CTR30, respectively (Fig. S1). Whole-genome sequencing revealed that *K. pneumoniae* strains belong to the K10-ST6262 lineage (Table 2).

**TABLE 1** MIC of *cfr*-positive *K. pneumoniae* strains J21CTR26, J21CTR30, and transconjugant 26-J53[a,b]

| Strain | MIC (mg/L) | | | | | | | | | | | | | | | | | |
|---|---|---|---|---|---|---|---|---|---|---|---|---|---|---|---|---|---|---|
| | AMP | CRO | CTX | CAZ | DOX | TIG | TET | KAN | AMI | STR | CIP | ENR | LZD | COL | MEM | FOS | LIN | FFC |
| J21CTR26 | >512 | >512 | 512 | 32 | 512 | 2 | >512 | >512 | >512 | 4 | 256 | 256 | >512 | 512 | 0.03 | >512 | >512 | 256 |
| J21CTR30 | >512 | >512 | >512 | 128 | >512 | 2 | >512 | >512 | >512 | 8 | 256 | >512 | >512 | 512 | 0.015 | >512 | >512 | 256 |
| 26-J53 | 512 | 0.015 | 0.015 | 0.03 | 32 | 0.25 | 128 | 0.25 | 0.25 | 1 | 0.015 | 0.015 | 128 | 1 | 0.015 | 1 | 64 | 512 |

[a]AMI, amikacin; AMP, ampicillin; CAZ, ceftazidime; CIP, ciprofloxacin; COL, colistin; CRO, ceftriaxone; CTX, cefotaxime; DOX, doxycycline; ENR, enrofloxacin; FFC, florfenicol; FOS, fosfomycin; KAN, kanamycin; LIN, lincomycin; LZD, linezolid; MEM, meropenem; STR, streptomycin; TET, tetracycline; TIG, tigecycline.
[b]Drug resistance is shown in bold black.

**TABLE 2** Genomic characteristics of *K. pneumoniae* strains J21CTR26 and J21CTR30[a]

| Strain | Serotype | Multilocus sequence typing | Plasmids | Self-transfera-bility | Size (bp) | Plasmid type | Resistance results | Virulence results |
|---|---|---|---|---|---|---|---|---|
| J21CTR26 | KL10 | ST6262 | Chromosome | | 5,404,868 | | $bla_{SHV-11}$, *fosA*, and *oqxA/B* | *fimH*, *mrkA*, and *iutA* |
| | | | pCTR26-1 | – | 485,030 | IncFII(K), IncFIB(K), IncHI2A, and IncHI2 | $bla_{CTX-M-65}$, *aac(3)-IV*, *aadA1/2/2b*, *aph(4)-Ia*, *fosA3*, *floR*, *erm(B)*, *mph(A)*, *dfrA12*, *sul2/3*, *cmlA1*, and *qacL* | *traT* and *terC* |
| | | | pCTR26-2 | Conjugative | 141,660 | IncFII(K), IncR, and IncFIB(pQil) | $bla_{TEM-1B}$, *cfr*, *floR*, *tet*(M), and *tet*(D) | *traT* |
| | | | pCTR26-3 | – | 106,770 | IncFII(K) | $bla_{DHA-1}$, *aph(3ʹ)-Ia*, *qnrB4*, *armA*, *sul1*, *mph(E)*, *msr(E)*, and *qacE* | *traT* |
| J21CTR30 | KL10 | ST6262 | Chromosome | | 5,405,166 | | $bla_{SHV-11}$, *fosA*, and *oqxA/B* | *fimH*, *mrkA*, and *iutA* |
| | | | pCTR30-1 | – | 394,261 | IncHI2A, IncHI2, IncFII(K), IncFIB(pQil), and IncR | $bla_{CTX-M-65}$, *aac(3)-IV*, *aadA1/2*, *aph(4)-Ia*, *fosA3*, *floR*, *erm(B)*, *mph(A)*, *dfrA12*, *sul2/3*, *cmlA1*, *qacL*, $bla_{TEM-1B}$, *cfr*, *tet*(M), and *tet*(D) | *terC* and *traT* |
| | | | pCTR30-2 | – | 234,624 | IncFII(K) and IncFIB(K) | None | *traT* and *terC* |
| | | | pCTR30-3 | – | 106,770 | IncFII(K) | $bla_{DHA-1}$, *aph(3ʹ)-Ia*, *qnrB4*, *armA*, *sul1*, *mph(E)*, *msr(E)*, and *qacE* | *traT* |

[a]"–" indicates uncertain conjugativity of the plasmid.

Strain J21CTR26 possessed a chromosome of 5,404,868 bp and three plasmids: pCTR26-1 (485,030 bp), a *cfr*-positive IncFII(K) plasmid pCTR26-2 (141,660 bp), and pCTR26-3 (106,823 bp). This strain harbored a total of 32 resistance genes, including $bla_{CTX-M-65}$, *fosA3*, *oqxA/B*, $bla_{SHV-11}$, and *cfr*, in addition to 5 virulence genes: *fimH*, *mrkA*, *iutA*, *traT*, and *terC*. Similarly, strain J21CTR30 had a chromosome of 5,405,166 bp and three plasmids: a *cfr*-positive multireplicon plasmid pCTR30-1 (394,261 bp), pCTR30-2 (234,624 bp), and pCTR30-3 (106,770 bp). It also carries 32 resistance genes and the same set of five virulence genes as J21CTR26 (Table 2). Conjugation experiments demonstrated that the *cfr*-positive plasmid pCTR26-2 can be transferred to the recipient strain *E. coli* J53 through conjugation, with a conjugation frequency of $4.2 \times 10^{-6}$ transconjugants per recipient. However, despite repeated attempts, no transconjugants were obtained for the plasmid pCTR30-1 in strain J21CTR30.

## Sequence analysis of plasmids

In the *K. pneumoniae* strain J21CTR26, plasmid pCTR26-1 is 485,030 bp in size with a guanine-cytosine (GC) content of 48.7% and is classified as a multireplicon plasmid of Inc FII(K)-IncFIB-IncHI2A-IncHI2 type. It comprises 363 ORFs including multiple resistance genes, such as $bla_{CTX-M-65}$, *fosA3*, *floR*, and *erm(B)*, among others. The segment spanning from 1 to 263,198 bp exhibited significant sequence similarity with the IncHI2 plasmid unnamed1 (accession number CP082955), which carried the $bla_{CTX-M-55}$ gene identified in clinical *Salmonella enterica* strains from Wuhan, China. This segment shared a conserved structural framework with unnamed1, including 13 tra genes (Fig. S2A). Additionally, the region from 262,053 to 485,029 bp demonstrated high sequence homology with the IncFII(K) plasmid pYTF44-1-233k (accession number CP075288), identified in *K. pneumoniae* isolated from pig manure in Jiangsu, China. Within pCTR26-1, two distinct multidrug resistance regions (MRRs) were identified, each flanked by IS*26* elements (Fig. S2B). The first MRR (MRR1), spanning from 190,008 to 209,774 bp, contains the *erm(B)*, $bla_{CTX-M-65}$, and *fosA3* genes, with a region of approximately 4,937 bp containing *erm(B)* resembling a fragment of the *Salmonella* plasmid unnamed1, and another part similar to a segment of the *fosA3*-harboring IncFII plasmid pESBL_DR28a found in *E. coli*. The second MRR (MRR2), spanning from 221,341 to 261,839 bp, contained the *aac(3)-IV*, *aph(4)-Ia*, *mph(A)*, *cmlA1*, and *floR* genes and exhibited 99.97% sequence

identity and 96% query coverage with the *mcr-1*-harboring IncHI2 plasmid pXGE1mcr found in a cattle manure-derived *E. coli* in Jiangsu, China. The segment located between 226,763 and 246,141 bp also showed 99.84% homology with the IncHI2 plasmid p573-1 from Zhejiang, China, with a query coverage of 86.34%. Compared to p573-1, the ISCR2 fragment is truncated and acquired the hp-*floR-virD2*. These observations suggested that pCTR26-1 possibly originated from the recombination events involving plasmids pYTF44-1-233k and unnamed1, likely entailing the insertion or excision of multiple genetic elements and resistance genes.

The plasmid pCTR26-2, carrying the *cfr* gene, is 141,660 bp in size, with a GC content of 51.4%. It encodes 342 ORFs and belongs to the multireplicon types IncFII(K)-IncR-IncFIB. Sequence analysis revealed that pCTR26-2 is a novel hybrid plasmid containing the backbone of an IncFII(K) plasmid, along with replicons and partial fragments from IncR and IncFIB plasmids, as well as additional gene fragments (Fig. 1A). It exhibited 96.16% homology with the *cfr*-negative IncFII(K) plasmid pKp1792_1 (CP085104.1, 151,942 bp) from a *K. pneumoniae* strain isolated from Norwegian shellfish, and the IncFII(K) plasmid pKp_SB611_3 (CP084846.1, 101,892 bp) from *K. pneumoniae* found in Dutch sewage but with lower coverage rates of 47% and 43%, respectively (Fig. 1A). Furthermore, pCTR26-2 contained two MRRs. The first MRR (MRR1), spanning from 87,628 to 105,375 bp, included *tet*(M) and *tet*(D) genes within a Tn*As1* composite class 4 integron, flanked by three copies of IS*26*. This transposon module, IS*26*-*tet*(D)-IS*26*-*tet*(M)-IS*5D*-IntI4-IS*26*, shared significant homology (71% coverage, 100% identity) with the *tet*(D)-containing region of the IncFII plasmid pSCLC9-2_3 carrying *bla*$_{OXA-1}$ from *K. pneumoniae* in Jiangsu, China. The second MRR (MRR2), located between 118,761 and 141,649 bp, contained *bla*$_{TEM-1B}$, *cfr*, and *floR* within a 19,017 bp transposon unit. This unit, ΔTn*3*-*bla*$_{TEM-1B}$-IS*1230B*-*repA*-IS*26*-*cfr*-IS*26*-IS*CR2*-*floR*-IS*26*-ΔTn*3*, was similar to the *floR*-containing segment of the IncR plasmid pTH114-1 from a clinical *K. pneumoniae* isolate in Thailand (55% coverage, 99.84% identity). The *cfr* gene is flanked by IS*26* elements, indicating mobilization possibly facilitated by IS26-mediated transposition, although no direct repeats were found at the IS*26* ends. Additionally, the 11,122 bp sequence between the two MRRs exhibited 99.84% homology with a segment of the IncR plasmid (pB16KP0089-1), including the replication protein gene from a clinical isolate of *K. pneumoniae* in South Korea, with a coverage of 100% (Fig. 1B). Evolutionary analysis indicated that IncFII(K) pKp_SB611_3 may represent an ancestral plasmid of pCTR26-2, with pCTR26-2 possibly evolving through the insertion of multiple genetic elements including fragments from IncR and IncFIB plasmids.

The plasmid pCTR26-3 has a size of 106,770 bp and a GC content of 51.6%. It comprises 366 ORFs and belongs to the IncFII(K) plasmid type. BLAST analysis showed that pCTR26-3 shared the highest homology with the IncFII(K) plasmid pHB25-1-88K (CP039525, 88,581 bp) from *K. pneumoniae* strain HB25-1, with a query coverage of 70% and 98.87% identity. Furthermore, pCTR26-3 exhibited 98.84% identity with the IncFII(K) plasmids pKP1161-1 and IMP-4_IncFI from *K. pneumoniae* strains isolated from sputum samples in Zhuhai and Taiwan, with coverage ranging from 81% to 84%. These plasmids shared a conserved backbone region essential for plasmid conjugation, indicating significant evolutionary relationships while displaying variation in MRR (Fig. S3). We propose that pHB25-1-88K may be an archetypal plasmid in the evolutionary lineage leading to pCTR26-3. The proposed formation process of pCTR26-3 involved the following steps: the insertion of a segment carrying *qnrB4* and *bla*$_{DHA-1}$ genes into pHB25-1-88K resulted in the formation of IMP-4_IncFI; subsequent insertion of a segment containing IS*kpn14* led to the formation of pKP1161-1; and finally, the insertion of an MRR carrying *msr(E)*, *mph(E)*, and *aph(3´)-Ia* genes resulted in the formation of pCTR26-3.

In the *K. pneumoniae* strain J21CTR30, the plasmid pCTR30-1 was a 394,261 bp hybrid plasmid carrying the *cfr* gene. Plasmid pCTR30-2 was a 234,624 bp non-resistant IncFII(K)-IncFIB(K) plasmid, and pCTR30-3 was identical to pCTR26-3 in strain J21CTR26.

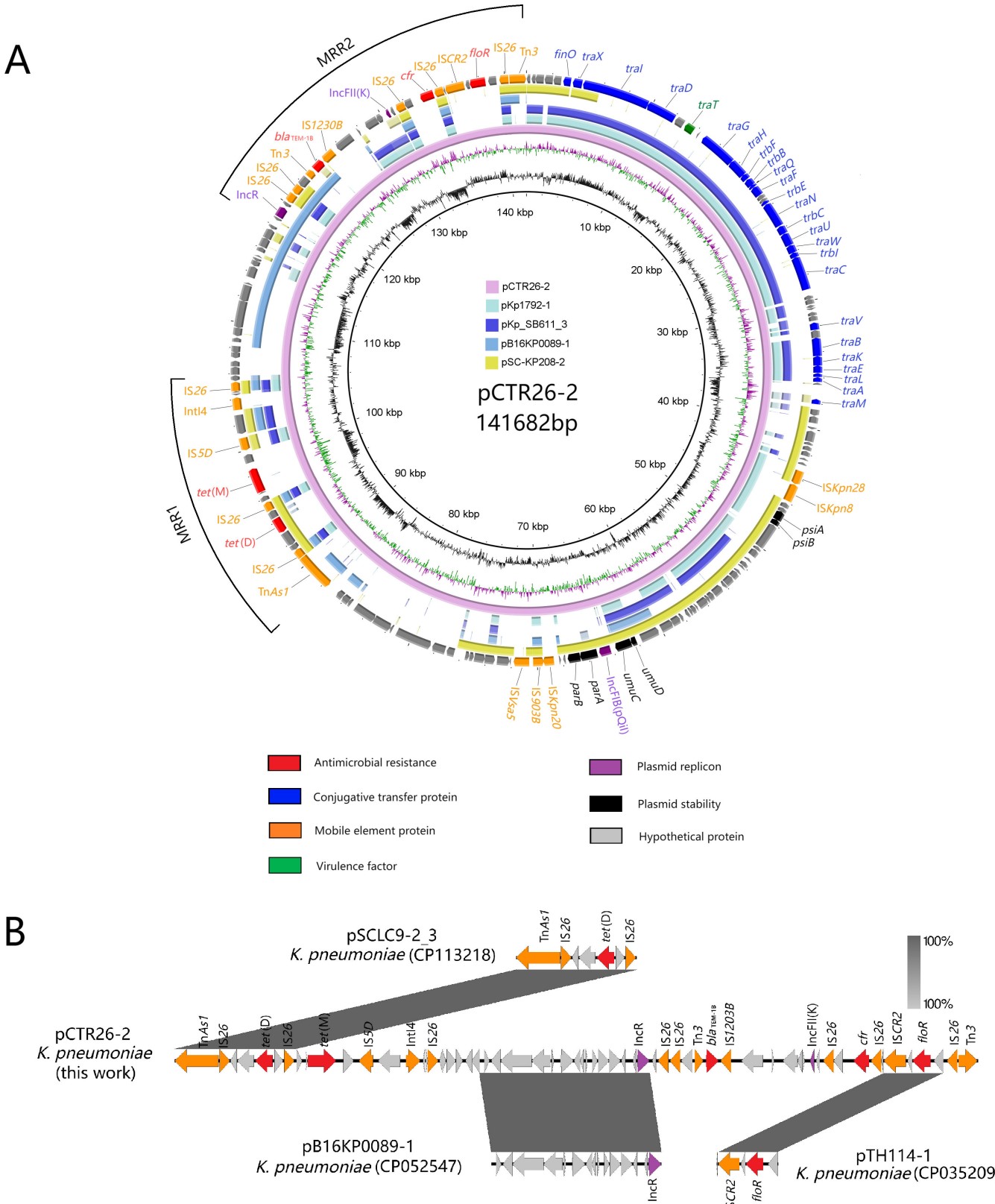

**FIG 1** Comparative analysis of the *cfr*-positive plasmid pCTR26-2 with similar sequences in GenBank. (A) Structural comparisons of the plasmid pCTR26-2 with similar plasmids in GenBank. (B) Comparison of the MRRs in pCTR26-2 with that of other plasmids. Blue, red, purple, orange, green, black, and gray arrows represent transfer-related proteins, resistance genes, replicons, mobile elements, virulence genes, plasmid stability proteins, and hypothetical and unclassified elements, respectively. Transcription direction is indicated.

Sequence analysis showed that pCTR30-1 had high homology with fragments of both pCTR26-1 and pCTR26-2 (Fig. S4).

## Fusion mechanism of pCTR30-1 mediated by Tn*As1*

Further analysis showed that the plasmid pCTR30-1 is a hybrid plasmid with chimeric characteristics mediated by Tn*As1*. The pCTR30-1 plasmid consisted of pCTR26-1 (nucleotide [nt] 1–110,444; 250,236–290,704; and 302,321–394,261) and pCTR26-2 (nt 110,445–250,235), missing a specific fragment containing IS*CR2-floR* from pCTR26-1, but acquiring an 11,617 bp segment containing *drfA12* and *aph(3')-Ia*. Two copies of Tn*As1*, oriented in the same direction, were embedded in the junction region of the fusion plasmid pCTR30-1.

Based on the sequence analysis and the observed structure, we proposed the model of pCTR30-1 formation shown in Fig. 2. In this model, the Tn*As1* element in pCTR26-1, after deleting a large fragment, interacted with another Tn*As1* in the conjugative plasmid pCTR26-2, resulting in the formation of a cointegrate. The linearized pCTR26-2 was incorporated into pCTR26-1, creating the cointegrate plasmid pCTR30-1, with the two Tn*As1* elements in the same orientation flanking the insertion fragment.

## Identification of the circular intermediate

Genomic investigations have identified a pair of identical IS*26* elements flanking the *cfr* gene in both *K. pneumoniae* strains J21CTR26 and J21CTR30. Reverse PCR and sequencing techniques revealed that these strains can generate an identical circular intermediate with a size of 3,142 bp, as illustrated in Fig. S5. This circular intermediate (IS*26-cfr*) was formed through the recombination of two identical IS*26* copies oriented in the same direction, potentially facilitating the transfer of the *cfr* gene via IS*26*-mediated recombination.

## DISCUSSION

The escalating issue of antimicrobial resistance (AMR), particularly multidrug resistance, is now recognized as a significant global public health threat in this century (16). The discovery of the multidrug-resistant gene *cfr* in a bovine-derived *Staphylococcus* isolate has spurred extensive research into its transmission routes and patterns. Initially, the *cfr* gene was predominantly identified in gram-positive bacteria such as *Staphylococcus* and *Enterococcus* spp. (17). However, it has progressively disseminated to gram-negative bacteria. In this study, two strains of XDR *K. pneumoniae* carrying the *cfr* gene were isolated from environmental samples at a poultry farm in Jiangxi Province. These strains also harbored multiple resistance genes, including *cfr*, *bla*$_{\text{CTX-M-65}}$, *fosA3*, *tet*(M), and *oqxA/oqxB*. Since the initial identification of *cfr*-carrying *K. pneumoniae* in pig feed (5), there has been limited knowledge regarding the prevalence of *cfr*-carrying *K. pneumoniae*. In this study, *cfr* was detected for the first time in a *K. pneumoniae* isolate from a chicken farm environment. The multidrug resistance of these strains indicates their potential to survive under the pressure of various drugs, such as cephalosporins, tetracyclines, quinolones, and others.

Mobile genetic elements, such as plasmids, insertion sequences, and transposons, play critical roles in the dissemination of the *cfr* gene across both gram-negative and gram-positive bacteria, with plasmids being particularly pivotal mediators of *cfr* transmission. Various *cfr*-carrying plasmids such as IncFII, IncX4, IncA, IncC, IncN, and IncP have been reported, yet no *cfr*-carrying IncFII(K)-type plasmid had been identified prior to this study. The prevalence of IncFII(K) plasmids in *K. pneumoniae* has become a significant concern due to their role in the dissemination of ARGs (18). IncFII(K) plasmids are known for their broad host range among gram-negative bacteria, facilitating horizontal gene transfer (19). These plasmids often carry multiple resistance determinants, including genes conferring resistance to beta-lactams, aminoglycosides, and quinolones, posing a significant threat to public health by limiting treatment

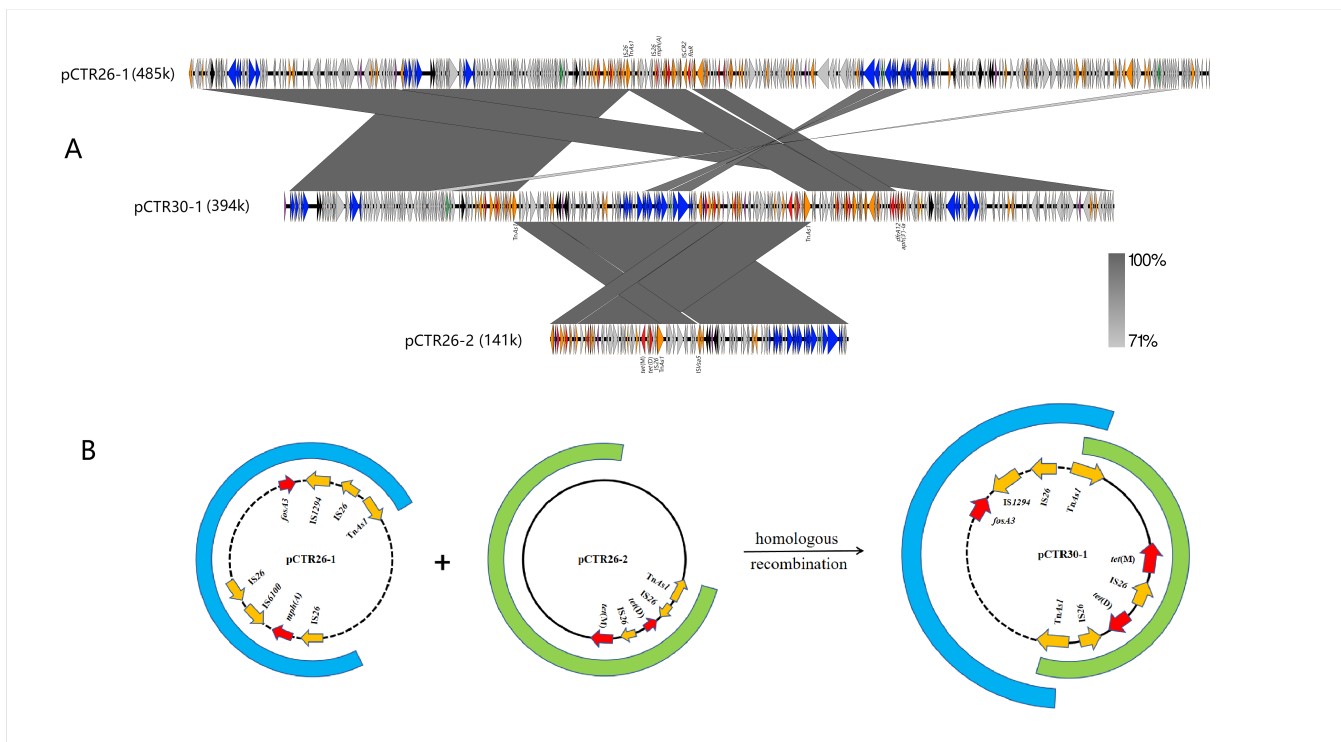

**FIG 2** Linear sequence comparison (A) and proposed fused model (B) of pCTR30-1, pCTR26-1, and pCTR26-2. Colored arrows indicate ORFs, and blue, orange, black, red, purple, green, and gray represent transfer-related genes, mobile elements, stability-related genes, resistance genes, replicon genes, virulence genes, and hypothesis proteins, respectively.

options for infections caused by *K. pneumoniae* (20). Recent studies have documented the widespread detection of IncFII(K) plasmids in clinical isolates of *K. pneumoniae* across various regions, underscoring their contribution to the global spread of multi-drug resistance (21, 22). This study identified two novel *cfr*-positive hybrid plasmids, pCTR26-2 and pCTR30-1, in two *K. pneumoniae* isolates, characterized by the insertion of the *cfr* gene flanked by IS*26* into the IncFII(K) plasmid backbone. This finding further demonstrated the ability of IncFII(K) plasmids to integrate and mobilize additional resistance genes through mechanisms like IS*26*-mediated transposition. The transfer of the *cfr* gene is facilitated by insertion sequences such as IS*26* or SXT-like integrative and conjugative elements, enabling its movement between different plasmids and bacterial chromosomes, contributing to the global spread of *cfr*-mediated resistance (23). The broad host range of these plasmids further amplified their impact. Understanding the epidemiology of *cfr*-carrying plasmids is crucial for developing effective strategies to combat the proliferation of multidrug-resistant (MDR) bacteria.

Conjugative plasmids play a crucial role in horizontal gene transfer via bacterial conjugation. Recent studies have increasingly highlighted the role of conjugative plasmids as facilitators in the transfer of non-conjugative MDR and virulence plasmids (24). The fusion of plasmids, achieved through replicative transposition or homologous recombination (HR), had been shown to broaden both the resistance and virulence profiles and the host range (25–28). For instance, IS*26*-mediated replicative transposition had integrated the *Salmonella*-specific virulence plasmid pSEN into the IncHI2 MDR plasmid (29). Similarly, IS*Pa40*-mediated homologous recombination in *Salmonella* had led to the formation of a fusion plasmid (30). In *Enterococcus*, fusion events occurred through IS*1216E*-mediated replicative transposition, connecting *poxtA*-positive plasmids to plg1-like plasmids or through HR within parent strain plasmids containing identical sequences (28). However, these fused plasmids were obtained under laboratory conditions through conjugation experiments, and their copy numbers were very low in

the donor strain. The genetic stability of these fusion plasmids in clinical strains remains to be confirmed. In this study, two *cfr*-positive plasmids, pCTR26-2 and pCTR30-1, were identified in *K. pneumoniae* isolates J21CTR26 and J21CTR30, respectively. Sequence analysis revealed that the pCTR30-1 plasmid resulted from the fusion of pCTR26-2 and pCTR26-1 through Tn*As1*-mediated homologous recombination. Parent and daughter plasmids are identified in two different strains, indicating that fusion plasmids possess stable heritability and transferability. Additionally, there are only 15 SNP differences between J21CTR26 and J21CTR30. According to the medication usage records, under the selective pressure of florfenicol in this farm, fusion plasmids are more likely to form and stably proliferate, promoting the accumulation and dissemination of resistance genes. Therefore, we should adopt a more prudent antimicrobial use strategy to reduce unnecessary drug abuse and avoid putting selection pressure on bacteria.

Horizontal gene transfer, mediated by mobile genetic elements, is a crucial mechanism driving the environmental dissemination of ARGs. Recent findings have highlighted that once carbapenemase-producing Enterobacterales contaminate an environment, these pathogens can become firmly established within farm settings (31). The environmental reservoir of AMR poses a significant risk as it can transition from agricultural environments to livestock, and subsequently, AMR traits in poultry can infiltrate the human food chain during processing, as suggested by Berglund (32). In this study, the identification of two XDR *K. pneumoniae* strains carrying novel *cfr*-positive plasmids, isolated from environmental samples at a chicken farm, highlighted the potential for these isolates and plasmids to spread across different ecological niches, thereby posing a significant threat to human health.

## Conclusion

In summary, the complete genetic features of the genome and plasmids in two XDR *K. pneumoniae* strains isolated from environmental samples at a chicken farm were analyzed. The present study is the first to detect the *cfr* gene in *K. pneumoniae* isolates from the chicken farm environment, with *cfr* located on novel plasmids. Notably, the *cfr*-positive plasmid pCTR30-1 in strain J21CTR30 resulted from the fusion of the *cfr*-positive plasmids pCTR26-2 and pCTR26-1 from strain J21CTR26, facilitated by Tn*As1*-mediated homologous recombination. This highlighted the dynamic recombination and spread of resistance plasmids in agricultural environments. Our findings provided insights into the dissemination and evolution of *cfr*-positive plasmids in husbandry, posing a serious threat to public health. Therefore, prudent use of antimicrobial agents, particularly antibiotic combinations, is crucial in clinical practice to prevent the occurrence, dissemination, and further evolution of MDR plasmids. Additionally, rigorous environmental disinfection practices are essential to control the spread of resistant bacteria.

## ACKNOWLEDGMENTS

This work was supported by the National Natural Science Foundation of China (no. 32072923), the Program for Science and Technology Innovation Talents in Universities of Henan Province (no. 23HASTIT039), the Henan Province Outstanding Youth Science Fund Project (242300421109), and the Science and Technology Innovation Fund project of Henan Agricultural University (2023CXZX009).

## AUTHOR AFFILIATIONS

[1]College of Veterinary Medicine, Henan Agricultural University, Zhengzhou, China
[2]Ministry of Education Key Laboratory for Animal Pathogens and Biosafety, Zhengzhou, Henan Province, China
[3]Henan Province Key Laboratory of Animal Food Pathogens Surveillance, Zhengzhou, China

## AUTHOR ORCIDs

Yiming Feng 🔴 http://orcid.org/0009-0008-7637-7070
Yajun Zhai 🔴 http://orcid.org/0009-0002-2218-4819
Li Yuan 🔴 http://orcid.org/0000-0001-5602-6910
Yushan Pan 🔴 http://orcid.org/0000-0001-8207-8941
Dandan He 🔴 http://orcid.org/0000-0002-1536-6465

## DATA AVAILABILITY

Data will be made available on request. The whole-genome sequencing data of strains J21CTR26 and J21CTR30 can be accessed through BioProject IDs PRJNA1134176 and PRJNA1134183. The sequences of plasmids pCTR26-1, pCTR26-2, pCTR26-3, pCTR30-1, pCTR30-2, and pCTR30-3 can be obtained through CP160465-CP160467 and CP160640-CP160642.

## ADDITIONAL FILES

The following material is available online.

### Supplemental Material

**Supplemental material (Spectrum02628-24-S0001.docx).** Fig. S1 to S5.

### Open Peer Review

**PEER REVIEW HISTORY (review-history.pdf).** An accounting of the reviewer comments and feedback.

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
