## [Reviewer comments · Microbiology Spectrum]

Microbiology Spectrum

Identification and evolution of novel *cfr*-carrying plasmids in XDR *Klebsiella pneumoniae* strains from a chicken farm

Yiming Feng, Tiantian Li, Shiyun Zhao, Mengxiang Zheng, Jiaying Shen, Xiaoying Wu, Xuexue Li, Ya-jun Zhai, Li Yuan, Jian-Hua Liu, Gongzheng Hu, yushan pan, and Dandan He

Corresponding Author(s): Dandan He, Henan Agricultural University

Review Timeline:

Submission Date:	January 3, 2025
Editorial Decision:	January 24, 2025
Revision Received:	January 27, 2025
Editorial Decision:	February 3, 2025
Revision Received:	February 5, 2025
Accepted:	February 26, 2025

Editor: Zhangqi Shen

Reviewer(s): The reviewers have opted to remain anonymous.

Transaction Report:

DOI: <https://doi.org/10.1128/spectrum.02628-24>

Re: Spectrum02628-24 (**Identification and evolution of novel *cf*r-carrying plasmids in XDR *Klebsiella pneumoniae* strains from a chicken farm**)

Dear Prof. Dandan He:

Thank you for the privilege of reviewing your work. Below you will find my comments, instructions from the Spectrum editorial office, and the reviewer comments.

Revision Guidelines

Sincerely,
Zhangqi Shen
Editor
Microbiology Spectrum

Re: Spectrum02628-24R1 (**Identification and evolution of novel *cfr*-carrying plasmids in XDR *Klebsiella pneumoniae* strains from a chicken farm**)

Dear Prof. Dandan He:

Thank you for the privilege of reviewing your work. Below you will find my comments, instructions from the Spectrum editorial office, and the reviewer comments.

This is a transferred manuscript. It seems that the author has addressed most of the questions raised by the reviewer. However, this manuscript needs further improvement. The author should avoid making quick and rushed conclusions. For example, in lines 325-327, the sentence, 'In addition, there were minor SNPs (15 SNPs) between J21CTR26 and J21CTR30 with the same ST-type, indicating that *cfr* was clonally transmitted in *K. pneumoniae*,' is incorrect. According to Table 2, although these two strains are similar, they do not belong to the same clone. Furthermore, the author cannot make such a conclusion without deciphering the other *cfr*-positive strains. Please read the whole manuscript and make the necessary modifications.

Revision Guidelines

Sincerely,
Zhangqi Shen
Editor
Microbiology Spectrum

Re: Spectrum02628-24R2 (**Identification and evolution of novel *cfr*-carrying plasmids in XDR *Klebsiella pneumoniae* strains from a chicken farm**)

Dear Prof. Dandan He:

Your manuscript has been accepted, and I am forwarding it to the ASM production staff for publication. Your paper will first be checked to make sure all elements meet the technical requirements. ASM staff will contact you if anything needs to be revised before copyediting and production can begin. Otherwise, you will be notified when your proofs are ready to be viewed.

Sincerely,
Zhangqi Shen
Editor
Microbiology Spectrum